# Molecular Changes in the Ischemic Brain as Non-Invasive Brain Stimulation Targets—TMS and tDCS Mechanisms, Therapeutic Challenges, and Combination Therapies

**DOI:** 10.3390/biomedicines12071560

**Published:** 2024-07-13

**Authors:** Aleksandra Markowska, Beata Tarnacka

**Affiliations:** Department of Rehabilitation Medicine, Faculty of Medicine, Warsaw Medical University, Spartańska 1, 02-637 Warsaw, Poland; beata.tarnacka@wum.edu.pl

**Keywords:** transcranial magnetic stimulation, transcranial direct current stimulation, ischemic stroke, neurorehabilitation, non-invasive brain stimulation, neuroinflammation, pyroptosis, ferroptosis, combination therapies, personalized medicine

## Abstract

Ischemic stroke is one of the leading causes of death and disability. As the currently used neurorehabilitation methods present several limitations, the ongoing research focuses on the use of non-invasive brain stimulation (NIBS) techniques such as transcranial magnetic stimulation (TMS) and transcranial direct current stimulation (tDCS). NIBS methods were demonstrated to modulate neural excitability and improve motor and cognitive functioning in neurodegenerative diseases. However, their mechanisms of action are not fully elucidated, and the clinical outcomes are often unpredictable. This review explores the molecular processes underlying the effects of TMS and tDCS in stroke rehabilitation, including oxidative stress reduction, cell death, stimulation of neurogenesis, and neuroprotective phenotypes of glial cells. A highlight is put on the newly emerging therapeutic targets, such as ferroptotic and pyroptotic pathways. In addition, the issue of interindividual variability is discussed, and the role of neuroimaging techniques is investigated to get closer to personalized medicine. Furthermore, translational challenges of NIBS techniques are analyzed, and limitations of current clinical trials are investigated. The paper concludes with suggestions for further neurorehabilitation stroke treatment, putting the focus on combination and personalized therapies, as well as novel protocols of brain stimulation techniques.

## 1. Introduction

### 1.1. Stroke Statistics and Current Rehabilitation Methods

Ischemic stroke is one of the major causes of death and disability. There are over 12.2 million new strokes each year. Globally, one in four people over age 25 will have a stroke in their lifetime. Six and a half million people die from stroke annually [1]. Around one in three stroke survivors will develop clinically significant symptoms of depression [2], 30% will develop anxiety [3], and over 70% will suffer from cognitive deficits [4]. As predicted in thirty-year projections of stroke epidemiology, the number of people living with stroke is estimated to increase by 27% between 2017 and 2047 in the European Union [5]. As the therapeutic time window for reperfusion therapy is narrow [6], significantly limiting the percentage of eligible patients [7,8], there is a tremendous need for effective stroke rehabilitation methods. The current options involve mainly physical therapy, speech therapy, and occupational therapy. However, they are time-consuming, require high patient compliance, and neuropsychiatric complications such as post-stroke depression may seriously reduce patients’ willingness to participate in rehabilitation and delay the recovery process [9,10,11]. Non-invasive brain stimulation (NIBS) is an increasingly used [12,13,14,15], novel, safe, and effective technique and may be a valuable alternative to conventional stroke rehabilitation methods. 

### 1.2. TMS and tDCS: Characteristics and Differences

Transcranial magnetic stimulation and transcranial direct current stimulation are two of the most popular non-invasive brain stimulation (NIBS) techniques [16]. They both modulate the excitability of the cortex and synaptic plasticity, being able to induce long-term potentiation (LTP) and long-term depression (LTD) changes, which contributes to the long-term effects of NIBS techniques, outlasting the period of stimulation [17,18,19]. However, there are differences that may prove important when choosing a particular one in research.

TMS uses a magnetic field to induce an electrical current sufficient to depolarize neurons and trigger action potentials in the stimulated cortical area [20,21]. It involves different modes, including repetitive transcranial stimulation (rTMS) and theta burst stimulation (TBS). rTMS involves delivering equal intervals of magnetic pulses of fixed frequency [22,23]. Low-frequency rTMS (LF-rTMS) (0–2 Hz) has been shown to have an inhibitory effect on cortex excitability, whereas high-frequency rTMS (HF-rTMS) (>5 Hz) has been observed to increase it [24,25]. HF-rTMS was found to improve the glutamatergic synaptic transmission in mouse peri-infarct cortex by regulating integrin α3/AMPA receptors signaling pathway and leading to motor function recovery after stroke [26]. Moreover, it was shown to increase dendritic complexity in mouse prefrontal and primary motor cortex, inducing intracortical rearrangement of neural circuits and modifying cortical connectivity [27,28]. Interestingly, apart from structural remodeling in the medial prefrontal cortex, HF-rTMS-treated mice also showed anti-depressant-like activity [28]. Another form of rTMS is theta burst stimulation, which involves the application of three bursts of high-frequency (50 Hz) pulses with an in-between interval of 200 ms that mimic the endogenous theta rhythms and induce LTP and LTD changes [24,25,29,30]. The key feature of TBS is the long-lasting effect of synaptic plasticity despite the short duration of the application period [31,32]; conventional rTMS sessions last up to 40 min, whereas TBS treatments take 1 to 3 min [25,33].

tDCS, on the other hand, uses weak direct electric current (1–2 mA) through the electrodes placed on distant regions of the scalp as anode and cathode. Contrary to TMS, it is not able to trigger the action potential but alters the resting membrane potential of neurons, changing the probability of discharge [16,21,34]. Analogously to TMS, anodal tDCS increases cortex excitability, and cathodal tDCS decreases it. Importantly, stroke causes a decrease in cortical excitability in the affected hemisphere and a compensatory increase in the unaffected hemisphere, which increases interhemispheric transcallosal inhibition and suppresses the activity in the lesioned hemisphere [20]. Therefore, applying different types of tDCS and TMS at different frequencies may exert an excitatory or inhibitory effect on the ischemic cortex. However, research on non-invasive treatment is mainly limited to the chronic phase of stroke [35,36], and brain stimulation techniques are being increasingly investigated in the early stages. In mice models, cathodal tDCS was found to reverse the maladaptive hyperconnectivity in the subacute phase of stroke and promote motor recovery [37]. Similarly, bihemispheric tDCS, with an anode placed over the lesioned motor cortex and a cathode over the contralateral one, applied in the early subacute phase, accelerated the rate of motor recovery in stroke mouse model [38]. As an important note, encouraging results from the use of tDCS in acute and subacute phases of stroke are pivotal for patients who are not able to start physiotherapy in the early stage. However, Peruzzotti-Jametti et al. demonstrated that, although cathodal stimulation applied in the acute phase of stroke preserved cortical neurons from the ischemic damage and decreased cortical glutamate, the anodal stimulation provoked an increase in lesion volume and blood–brain barrier disruption, which may be the result of early hyperreperfusion and hyperemia [36].

Both TMS and tDCS show effectiveness in functional recovery after stroke, and until now, neither technique has been proven to be more beneficial in stroke rehabilitation. However, there are very few studies comparing their outcomes [39], or they have small sample sizes [40], which affects the significance of the results. Notwithstanding, the differences in their mechanisms of action can prove important in the selection of the most appropriate approach for specific therapeutic goals. As TMS induces a more focal electrical field than tDCS and generates action potentials in specific neural circuits, it is presumed to be more effective in stimulating specific white matter tracts and affecting transcallosal neurons than tDCS [41]. Interestingly, the application of LF-rTMS over contralesional M1 was found to be more beneficial in fine hand movement than for large motor function [40]. As the fine movements require greater engagement of the cortical networks of contralateral M1, higher interhemispheric interaction from the non-lesional hemisphere may inhibit the engagement of the lesioned M1, which may, in turn, negatively affect motor recovery in stroke patients [42]. Therefore, applying LF-rTMS over non-lesioned M1, which decreases high interhemispheric inhibition from the contralesional M1, may improve fine movements. Similarly, bihemispheric tDCS has been demonstrated to modulate intracortical inhibitory pathways in the contralesional primary motor cortex [43] and modulate plasticity within ipsilesional and contralesional motor cortices, leading to reorganization of interhemispheric interactions [44]. Both TMS and tDCS have been found to decrease neuropathic pain, including central post-stroke pain, with rTMS having a slightly superior level of pain relief [45]. In a comparative study of TMS and tDCS on cognitive recovery after stroke, HF-rTMS was found to be the most promising therapeutic option in enhancing global cognitive function, particularly over the left dorsolateral prefrontal cortex (DLPFC). Moreover, dual-tDCS over bilateral DLPFC was shown to be superior to other NIBS in improving memory function. On the other hand, the effects of both stimulation techniques on attention, executive function, and activities of daily living (ADL) were found to be insignificant [46]. A different network meta-analysis confirms that DLPFC is the most promising target for cognitive recovery using NIBS techniques and recognizes HF-rTMS as the superior NIBS technique for ameliorating cognitive impairment. Nevertheless, dual-rTMS was found to be most effective in improving ADL functioning and LF-rTMS in alleviating unilateral spatial neglect [47].

In comparison to large and heavy TMS equipment, tDCS devices are simple, small, and portable and can be used at home [Table 1]. Their costs are also lower than those of TMS devices. However, their disadvantages include low spatial resolution and difficulty in precisely localizing the electric field current [48]. TMS, on the other hand, is characterized by high temporal and spatial resolution, which allows for targeting specific neural circuits [49]. However, they are more expensive, are not portable, and do not allow for home therapy. Possible adverse effects of TMS include seizures and syncope [48]. tDCS has been associated with fatigue, headache, skin redness, itching, and burning sensation under the stimulation electrodes [48]. Both techniques mostly target cortical regions and cannot stimulate subcortical areas without affecting the cortex [48].

As NIBS techniques outlast the period of stimulation [52], a variety of processes affecting synaptic plasticity, cell viability, and neurogenesis must occur. In the next chapters, a detailed analysis of the effect of NIBS on cellular and molecular mechanisms will be presented. Neuroinflammation, oxidative stress, apoptosis, pyroptosis, ferroptosis, and neurogenesis will be explored.

## 2. Discussion

### 2.1. Cell Death: Apoptosis, Pyroptosis, Ferroptosis, Necroptosis

The mitochondrial dysfunction and oxidative stress that follow brain ischemic injury lead to apoptotic and non-apoptotic programmed cell death [53,54,55]. After the TLR4/DR (toll-like receptor/death receptor) on the cell membrane receives the inflammatory signal, it triggers apoptosis, pyroptosis, necroptosis, and ferroptosis [56,57].

Contrary to necrosis occurring immediately in the core of ischemic injury, apoptosis occurs in the penumbra zone, around the core, within several hours or days [58,59,60]. In the ischemic brain, two main apoptotic pathways can be activated. In the intrinsic pathway, pro-apoptotic Bcl-2 (B-cell lymphoma 2) family proteins Bax (Bcl-2-associated protein X) and Bak (Bcl-2 homologous antagonist/killer) form pores in the outer mitochondrial membrane which results in the release of intermembrane space proteins such as cytochrome c and apoptosis-inducing factor (AIF) into cytosol which in turn activates the caspases (caspase-3, caspase-9) that cause DNA fragmentation, chromatin condensation and cell destruction [53,54,61]. It is reported that the balance between Bcl-2 and Bax plays a key role in apoptotic mechanisms determining whether the cell will survive or undergo programmed death [55]. Importantly, the decreased Bcl-2/Bax ratio is described in animal stroke models [62]. In the extrinsic pathway, extracellular ligands (such as tumor necrosis factor (TNF)-α, Fas ligand, TRAIL, glucocorticoids) bind to the death receptors on the cell membrane and activate the intracellular caspase-8 and caspase-3 leading to apoptotic cell death [53,54,61].

Importantly, Guo et al. demonstrated that rTMS significantly increased the expression of Bcl-2 and decreased the expression of Bax. Moreover, TUNEL staining detecting DNA breakage during early and late stages of apoptosis [63] showed that rTMS downregulated neuronal apoptosis [64]. Similarly, tDCS treatment was found to increase the Bcl-2/Bax ratio in MCAO rats as well as decrease the caspase-3 level [65]. Furthermore, Zong et al. reported that rTMS inhibited the intrinsic mitochondrial caspase-9/3 apoptotic pathway and attenuated delayed apoptotic cell death in the peri-infarct area in the photothrombotic (PT) rat model of ischemic stroke [66]. Zhou et al. demonstrated that tDCS upregulated BDNF–TrkB and its downstream PI3K/Akt signaling, which antagonizes the pro-apoptotic activity of the Bcl-2 family, such as Bad and Bax [67], eventually protecting neurons from apoptosis [68]. 

Pyroptosis is a recently discovered pro-inflammatory form of programmed cell death that is triggered by inflammasome induction and mediated by gasdermin (GSDM) family proteins [69,70]. The inflammasome converts procaspase-1 into caspase-1, which cleaves GSDM, releasing an N-terminal fragment (N-GSDM) [56,71]. N-GSDM binds to the cell membrane and perforates it, which eventually leads to the cell rupture and release of pro-inflammatory substances (IL-1β, IL-18). Released inflammatory factors contribute to the pyroptosis of neural cells, glial cells, and endothelial cells, which results in blood–brain barrier (BBB) disruption and irreversible brain damage [56,72]. Luo et al. showed that theta-burst rTMS (iTBS) inhibited the expression of proteins associated with pyroptosis, such as caspase-1, IL-1β, IL-18, ASC, GSDMD, and NLRP1 in the peri-infarcted area and inhibited TLR4/NFκB/NLRP3 signaling pathway modulating microglial activation and inhibiting neuronal pyroptosis [73].

Ferroptosis is an iron-dependent programmed cell death. In an ischemic brain, microglia release substantial amounts of metalloproteinases that disrupt the extracellular matrix and dysregulate the blood–brain barrier. Increased BBB permeability results in iron ions influx into brain tissue and iron overload [74]. Iron is involved in a number of detrimental processes occurring after ischemia, i.e., release of free radicals, excitotoxicity, and inflammatory response [75]. Iron-dependent lipid peroxidation that takes place during ischemic stroke results in oxidative membrane damage and, consequently, cell death [76,77]. The inactivation of glutathione peroxidase 4 (GPX4) that converts toxic lipid hydroperoxides into non-toxic lipid alcohols, protecting against membrane lipid peroxidation, leads to ferroptotic cell death, and its expression tends to be decreased in animal ischemic models [78]. Interestingly, Zhou et al. found that HF-rTMS increased GPX4 levels and decreased ASCL4 and TFRC, key components of ferroptotic processes reversing the reduced GPX4 level and the elevated ASCL4 and TFRC levels in MCAO rats. In addition, it reduced the concentrations of pro-inflammatory factors such as IL-1β, IL-6, and TNF-α in the cerebrospinal fluid [79]. Similarly, Shen et al. observed that intermittent theta burst stimulation increased GPX4 levels [80]. 

### 2.2. Oxidative Stress

Lipid degradation of the cell membrane and mitochondrial dysfunction lead to excessive production of free radicals, which cause damage to DNA structure, protein denaturation, and lipid peroxidation [81], thereby causing cell death. Nicotinamide adenine dinucleotide phosphate (NADPH) oxidase is the main source of oxygen-reactive species [82]. Following ischemic stroke, its expression rapidly increases in neurons and brain vessels [83]. In experimental stroke models, it has been demonstrated that blocking NADPH activation protects against focal ischemic injury, reducing superoxide generation and improving neurological functioning [82,84]. Following cerebral ischemia, the main mechanism by which ischemic brain damage is reduced is through superoxide dismutase (SOD) enzymes that eliminate O_2_^−^ by converting it into H_2_O_2_ and O_2_ [85]. Manganese superoxide dismutase (MnSOD) is one of the most important antioxidant cell components [86]. MnSOD deficiency exacerbates cerebral infarction, and the reperfusion after cerebral ischemia reduces the expression of MnSOD [85]. Therefore, increasing its concentration, and thereby, neuroprotective effects, has become the subject of current research.

It was demonstrated that rTMS administration reduced NADPH oxidase activation and superoxide production in the peri-infarct cortical region in the photothrombotic stroke model and increased the MnSOD production, which attenuated oxidative neuronal damage measured by labeling lipid peroxidation, DNA double-strand breaks, and oxidized DNA damage [66]. Administrating tDCS caused an increase in SOD levels in the cerebral ischemia/reperfusion (I/R) model, especially in the c/a-tDCS mode, which is a combination mode of the cathodal current in the ischemia stage and anodal current in the reperfusion period [87].

### 2.3. Glial Cells

Although it has been previously thought that brain stimulation techniques primarily affect nerve cells [88], glial cells have now become a critical NIBS target.

It was demonstrated that TMS and tDCS can affect the morphology and activation of astrocytes and microglia. In response to tDCS, astrocytes displayed elongated cell bodies with cellular filopodia that were oriented perpendicularly to the direct current electric field [89]. Studies on deep brain stimulation suggest that the stimulation of astrocytes can set in motion the release of gliotransmitters, which can trigger axonal activation [90]. Moreover, tDCS was also shown to enlarge microglial soma size in an adrenergic receptor-dependent manner [91]. However, the role of soma enlargement in positive outcomes of tDCS is not fully understood. Microglial soma enlargement has been associated with inflammatory processes [92], and it is suggested that the pro-inflammatory molecules released by microglia may be involved in synaptic plasticity induced by tDCS [91].

In in vitro models of ischemia, high-frequency repetitive magnetic stimulation (HF-rMS) was shown to have a direct modulatory effect on astrocytes and stimulate the release of trophic factors, including GDNF and PDGF-BB from their secretome, which promoted neuronal survival after ischemic period [93]. Importantly, the presence of astrocytes was shown to be crucial to the beneficial effects induced by HF-rMS after ischemia [94]. tDCS was demonstrated to directly modulate gene expression in astrocytes upregulating *BDNF*, playing a key role in neuronal plasticity, survival, and growth [95], as well as *FOS*, the marker of cell activation, differentiation, and proliferation in isolated astrocytes in vitro [96]. Moreover, DCS was shown to promote microglial phagocytic activity, responsible not only for debris clearance but also network remodeling and engulfment of excess synapses, being a sign of neuroplasticity [97,98,99]. Interestingly, microglia can respond to DCS indirectly through neuron–microglia communication as well as can directly perceive weak electrical fields [98].

In the study on rats with middle cerebral artery occlusion (MCAO), rTMS administration inhibited the neurotoxic polarization of astrocytes, maintaining their neuroprotective phenotype. It was shown that rTMS reduced the expression of neurotoxic markers (iNOS), increased the expression of neuroprotective markers (arginase 1), promoted astrocytic synaptic formation, and alleviated neuronal apoptosis, which eventually promoted neurological functional recovery in vivo [100]. Similarly, tDCS was also found to positively influence the recovery of function. Administrating cathodal-tDCS (c-tDCS) in mice with PT stroke was demonstrated to enhance the ramification of microglia at the perilesional region and modulate the phenotype of microglia, shifting its activation towards anti-inflammatory response, indicated by higher expression of anti-inflammatory markers in the ischemic core [101]. Similarly, Walter et al. showed that tDCS received daily by mice with experimental focal cerebral ischemia enhanced neurogenesis in the subventricular zone, diminished microglia polarization toward the neurotoxic CD16/32+ M1 phenotype and stabilized microglia polarization toward neuroprotective CD206+ M2-phenotype [102]. Braun et al. found that c-tDCS promoted the recruitment of oligodendrocyte precursors towards the lesion; however, contrary to the current literature findings on the effects of NIBS on microglia polarization [101,102,103,104], they observed that c-tDCS promoted microglia M1 pro-inflammatory phenotype [105] which suggests that spatiotemporal dynamics of microglia are much more complex and at different phases of ischemic stroke the tDCS effect may differ. Single-cell RNA sequencing and cell–cell communication analysis confirm that within the early stage of acute ischemic stroke, microglia exhibit distinct heterogeneity rather than M1/M2 polarization [106].

### 2.4. Neurogenesis

Neurogenesis involves the proliferation of neural stem cells, migration of neuroblasts to the infarct zone, and differentiation into neurons [107]. In an adult brain, neurogenesis takes place primarily in two regions: in the subventricular zone (SVZ) located along the lateral ventricles and in the subgranular zone of the dentate gyrus [107,108]. Ischemia can trigger neurogenesis in an adult brain; however, it is insufficient to restore brain function after a stroke [109]. In the ischemic brain, the survival of new neurons is reduced due to the lack of neurotrophic factors and chronic inflammation [110]. Stroke-induced hypoxia was reported to increase Notch signaling in neural stem cells (NSCs), which initiated an irreversible switch from neurogenesis to gliosis [111,112], hindering brain repair by engulfing synapses [113].

In animal stroke models, non-invasive brain stimulation techniques have been shown to facilitate endogenous neural stem cell regeneration. Guo et al. demonstrated that rTMS upregulated the BDNF signaling pathway and promoted neurogenesis as well as suppressed apoptosis in the ipsilateral hippocampus of adult rats with cerebral focal ischemia [64]. Luo et al. found that rTMS promoted the proliferation of neural stem cells in the ischemic penumbra through a Ca2+ influx-dependent phosphorylated AKT/glycogen synthase kinase 3β/β-catenin signaling pathway [114]. Zong et al. reported that continuous theta-burst stimulation (cTBS), a modality of transcranial magnetic stimulation (TMS), significantly expanded the pool of neural progenitor cells and newly generated immature neurons, attenuated their apoptotic death and maintained their survival in the peri-infarct region in a photothrombotic stroke rat model [115]. Furthermore, Peng et al. demonstrated that the combination of human neural stem cells (hNSCs) transplantation and rTMS in a middle cerebral artery occlusion (MCAO) rat model accelerated the functional recovery after ischemic stroke. rTMS promoted the neural differentiation of hNSCs after transplantation in rats, and the combined therapy synergistically enhanced neurogenesis in the SVZ through the BDNF-TrkB signaling pathway and increased the expression of neurotrophin BDNF [116].

Studies using tDCS demonstrated similar outcomes. Lei et al. showed that bilateral tDCS promoted the migration of NSC-derived neuroblasts from SVZ toward the cathode direction into the post-stroke striatum, protects against neuronal death, and improves the functional recovery of rats subjected to ischemia-reperfusion injury [117]. Zhang et al. found that tDCS promoted the proliferation of NSCs in the subventricular zone in the MCAO rat model, accelerated NSC migration from the SVZ to the ischemic site, and promoted NSC differentiation to oligodendrocytes and neurons by inhibiting Notch 1 signaling pathway [118]. Braun et al. demonstrated that tDCS, independently of polarity, increased the area covered by neuroblasts in the SVZ of the ipsilateral hemisphere but had no effect on the dentate gyrus of the hippocampus or the contralateral hemisphere [105]. Pikhovych et al. found that cathodal, more than anodal, tDCS induced neurogenesis in the mouse brain [119].

Importantly, applying transcranial magnetic stimulation on the stroke hemisphere of patients with sub-acute stroke was demonstrated to modulate endogenous neuroplasticity. Plasma levels of neurogenesis and axonogenesis biomarkers such as miR-25 and netrin-1 were significantly increased in the rTMS-treated group [120]. Netrin-1 facilitates synaptic formation and axonal regeneration, and miR-25 promotes adult neural stem cell proliferation. However, it was also found to downregulate the level of BDNF, contradicting the results of the aforementioned studies. Interestingly, the level of BDNF was found to be both increased [64,116,121,122] and decreased [123,124] in the research investigating the effect of TMS, which suggests that different frequencies and durations of this technique, as well as population characteristics, may play a pivotal role in the final plasma BDNF levels [124]. Nonetheless, the outcome of the study showed an association between rTMS and neurogenesis/axonogenesis biomarker enhancement, indicating that HF-rTMS may modulate endogenous neurogenesis and axonal sprouting after ischemic stroke in humans.

### 2.5. Combination Therapies

As the TMS and tDCS have been shown to enhance endogenous neural stem cell proliferation, migration, survival, and differentiation [114,115,116], NIBS techniques are increasingly used in combination with stem cell transplantations in animal models to improve the differentiation of exogenous stem cells into mature neurons and improve their integration into neural networks in the lesioned brain. Combining TMS with BMSCs displayed a more efficient recovery in rats with spine injury in comparison with monotherapy by reducing neuronal apoptosis, increasing neurotrophic expression levels (GAP-43, NGF, BDNF), and downregulating the expression of glial fibrillary acidic protein (GFAP) [125], the marker of astrocytic activation mediating glial scar formation which is also released into the bloodstream after brain tissue damage and is associated with stroke severity [126,127]. Similarly, combination of rTMS with hMSCs in Parkinson’s disease rat model was shown to create a favorable microenvironment by elevating the expression of neurotrophic factors (BDNF, GDNF, NGF, PDGF), enhancing the expression of pro-inflammatory cytokines (IL-10) and suppressing pro-inflammatory cytokines (TNF-α, IFN-γ) to amplify immune modulation effects in a synergistic manner and the combination treatment was more effective than monotherapies [128]. The mechanisms investigated in the aforementioned neurological disorders, such as preventing glial scar formation, neuroinflammation, and cell death, as well as enhancing the expression of neurotrophic factors, are also important in stroke treatment. So far, the combination therapy in stroke animal models involved the use of human neural stem cells (hNSCs) in rats after ischemic stroke. rTMS was shown to promote the neural differentiation of hNSCs after transplantation in rats, and the combined therapy synergistically enhanced neurogenesis in the SVZ through the BDNF-TrkB signaling pathway and increased the expression of neurotrophin BDNF [116]. Most importantly, combination therapy has also demonstrated improvements in clinical outcomes in a stroke patient. A case report involving administrating MSCs and rTMS following the acute phase of the ischemic stroke has been shown to improve patient’s motor strength and cognitive functions, which was assessed by the National Institutes of Health Stroke Scale (NIHSS), Fugl-Meyer Assessment and Montreal Cognitive Assessment-Indonesian version (MoCA-INA) [129].

### 2.6. Personalised Therapy

Although non-invasive brain stimulation techniques have been increasingly used in stroke rehabilitation research, there is considerable variability in their clinical effects, often making their outcomes unpredictable. The source of variability can be divided into two groups; the first one is the result of different stimulation protocols and inconsistent methodology of studies, and the second one is caused by interindividual variations such as brain anatomy, connectivity, cortical excitability, severity of stroke lesions, extent of corticospinal tract damage, baseline level of function, neurochemistry, genetics and age [130] [Figure 1]. Quantification of these factors can be achieved through neuroimaging techniques such as mainly MRI and its modalities, including fractional anisotropy (FA) of diffusion tensor imaging (DTI), structural MRI (sMRI), and functional MRI (fMRI) [131]. A study analyzing motor networks in responder and non-responder groups using fMRI showed that responders to rTMS and tDCS presented an increased involvement of contralesional M1, greater interhemispheric connectivity, and higher motor network efficiency before the stimulation [132]. The outcome suggests that brain stimulation techniques may be effective in patients with disrupted network balance but with functional interhemispheric connectivity. Similarly, the integrity of interhemispheric connections was shown to be crucial in post-stroke neglect recovery and response to cTBS [133]. The stroke severity, depending on the extent of white matter tract damage, was also found to be a predictor of NIBS effectiveness [131,134,135,136]. tDCS was demonstrated to improve limb control for patients with mild impairment and worsen it for patients with moderate to severe impairment [134]. Fractional anisotropy was used to measure the asymmetry between the posterior limbs of the internal capsule, determining the CST integrity, making the neuroimaging techniques an important tool in identifying non-responder groups. Moreover, gene polymorphisms may also affect the neuroplastic response to brain stimulation techniques in stroke patients. Val66Val carriers of the *BDNF* gene showed a decrease in cortical excitability, while Val66Met carriers presented an initial increase in cortical excitability followed by delayed inhibitory response 30 min after the stimulation [137]. Val66Met polymorphism was associated with less improvement of post-stroke aphasia after cTBS treatment in comparison with Val66Val, which suggests that genotype plays an important role in responsiveness to brain stimulation [138]. Current clinical studies focus on personalized tDCS using individual electrical field models [139,140]. As the brain and cranial structure affect the distribution of current density [130], simulation models based on anatomical data are created to investigate optimal current parameters. Individual electrical field models are researched to personalize the current strength and placement of the electrode grid on the scalp [139]. New software is being developed to analyze magnetic resonance images of stroke patients to generate brain models and calculate the magnitude of the electric field generated by tDCS. Importantly, lesion location and brain atrophy are taken into consideration in order to maximize the effects of tDCS [140].

## 3. Conclusion, Limitations, and Directions for Further Research

Although applying transcranial magnetic stimulation and transcranial direct current stimulation have been associated with various neuroprotective and neuroregenerative effects, still many challenges of non-invasive brain stimulation therapies need to be overcome.

There is a large heterogeneity of stimulation parameters in the studies, which makes it difficult to accurately compare the results in meta-analyses and further apply them in clinical practice. Developing protocols with exact parameters involving optimal frequencies, spatial distribution, pulse numbers, stimulation time, and intervals personalized to different groups of patients need to be established in order to ensure their maximum safety and efficacy.

The majority of studies involve in vitro or animal models, which present several translational challenges. In vitro studies cannot accurately mirror the complex brain environments in the ischemic brain. As an example, the application of the electrical current across the monolayer of astrocytes in vitro may have different effects than across the brain [96]. Moreover, the intricate interactions between glial cells, neural cells, and endothelial cells cannot be properly investigated in vitro. Furthermore, in vivo studies apply rTMS coils made for humans to animals, and different head-to-coil size ratios may lead to reduced stimulation efficiency [141]. Importantly, the brain structure of rodents, used most frequently in stroke rehabilitation research, is different from that of humans. Mice and rats have smooth brain structures [142] with a different geometry than the folded human cortex, which may affect the properties of the electric field of rTMS [141]. In addition, in the research on microglia polarization, it is important to note that the microglia distribution profile is different in the human and rodent brains [143]. The translational issues and limitations of current clinical studies make it imperative to conduct more randomized controlled trials.

What is important from the clinical point of view is that current trials have short treatment duration [144], involve small samples not reaching statistical significance [145,146,147], and are not followed by long-term assessments to investigate permanent changes in the brain after the end of the stimulation period [144,147,148]. The results are mixed, showing as well no significant changes in functional connectivity [149]. Furthermore, there are very limited dose comparison studies, and largely varying stimulation doses between research papers may count for the mixed outcomes [150]. Meta-analysis and meta-regression results suggest a dose–response relationship with electrode size, charge density, and current density, although the lack of detailed dosages and administration methods in the analyzed studies were the main limitations [150]. Moreover, they lack neurophysiological studies exploring the neuroplasticity mechanisms and assessing the clinical improvement after the intervention [147]. Importantly, publication bias [151] and low quality of studies [152] with methodological limitations [153] of NIBS clinical trials are common outcomes of current meta-analyses, leaving the most effective stimulation mode still to be determined [154].

The underlying mechanisms of not only non-invasive brain stimulation techniques but also molecular changes occurring in the ischemic brain are still not fully elucidated, and more research should focus on newly emerging therapeutic targets such as ferroptotic and pyroptotic cell death.

In addition, as novel protocols of TMS, such as intermittent (iTBS) and continuous theta burst stimulation (cTBS), present several advantages over standard TMS, i.e., long-lasting effects despite short administration periods, more research should focus on their clinical applications.

Noteworthy are the numerous studies indicating the synergistic effects of combining brain stimulation techniques with other neuroregenerative and neurorehabilitation strategies [Figure 2]. Encouraging results have been obtained using the combined therapy with stem cells transplantations [116,128,129], nanomaterials such as superparamagnetic iron oxide nanoparticles (SPIONs) [155], acupuncture [156], pharmacotherapy [157,158,159], botulinum toxin [160,161], physical exercise and movement therapy [162,163,164,165,166,167,168], virtual reality [165,169,170], working memory tasks [171], mindfulness-based stress reduction [160], occupational therapy [161,172,173], music therapy [174], as well as combining TMS with tDCS [175].

Although more research is still necessary to link the therapeutic effects of NIBS with molecular mechanisms occurring in the ischemic brain, recent progress in preclinical and clinical studies, and growing understanding of factors affecting responsiveness to brain stimulation techniques provide a promising basis for finding an effective form of stroke rehabilitation method.

## Figures and Tables

**Figure 1 biomedicines-12-01560-f001:**
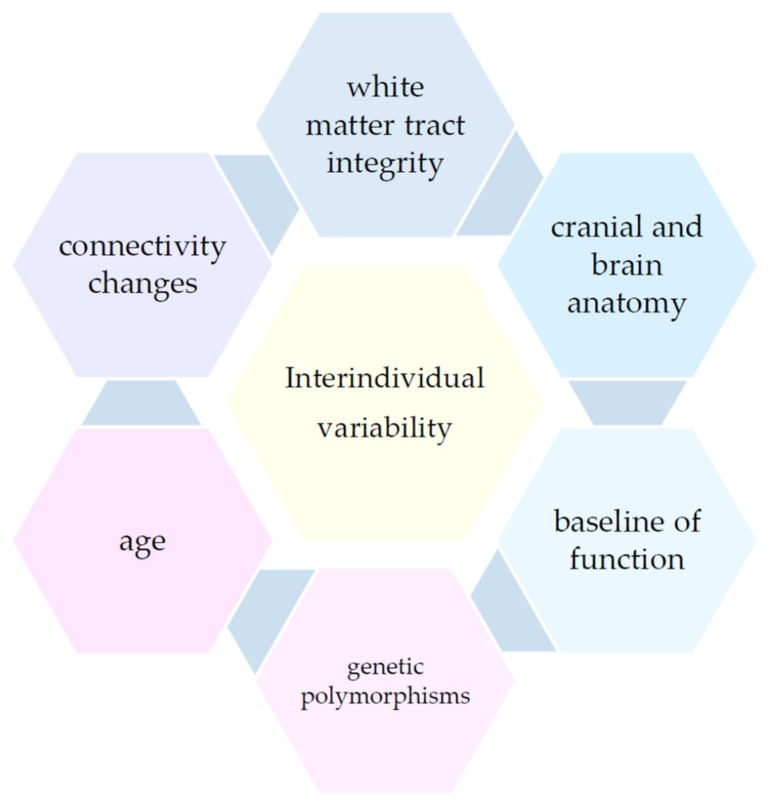
Interindividual variability in non-invasive brain stimulation effects.

**Figure 2 biomedicines-12-01560-f002:**
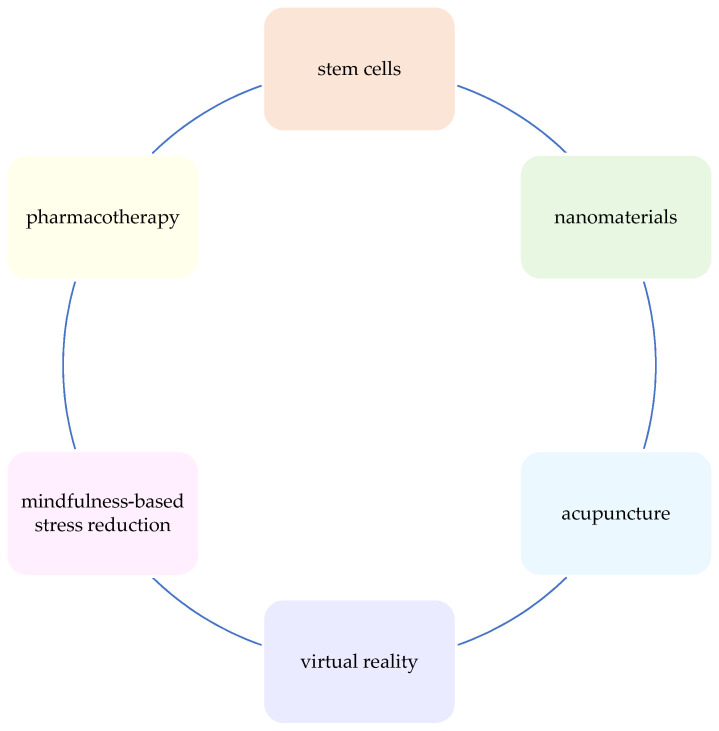
Combination therapies used with NIBS in stroke rehabilitation research.

**Table 1 biomedicines-12-01560-t001:** Characteristics and differences between TMS and tDCS.

	TMS	tDCS
Characteristics	Uses a magnetic field to induce electrical currents in the brain to modulate the excitability of the cortex	Delivers weak direct electric current (1–2 mA) through the electrodes placed on the scalp as anode and cathode to modulate the excitability of the cortex
Mechanism of interhemispheric modulation	Induces more focal electrical field and generates action potentials in a specific neural circuit [40]	Causes weak polarization of a larger number of neurons, which modulates synaptic activity during motor activation [40]
Possible adverse effects	Seizure and syncope [48]	Fatigue, headache, skin redness, itching, and burning sensation under the stimulation electrodes [48]
Size and portability of TMS and tDCS devices	Large, heavy, not portable [49]	Light, small, portable, and can be used at home [49]
Power supply requirements	Requires power supply	Battery driven
Costs	Higher cost (up to around USD 80,000) [50]	Lower cost (from around USD 100 to thousands of dollars) [51]
Neurophysiologic specificity	High temporal and spatial resolution allows for targeting specific neural circuits [49]	Low spatial resolution and difficulty in precisely localizing the electric field current [48]
Target regions	Mostly targets cortical regions and cannot stimulate subcortical areas without affecting the cortex [48]	Mostly targets cortical regions and cannot stimulate subcortical areas without affecting the cortex [48]

## Data Availability

Not applicable.

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
