# Peer review of "Molecular Changes in the Ischemic Brain as Non-Invasive Brain Stimulation Targets—TMS and tDCS Mechanisms, Therapeutic Challenges, and Combination Therapies"

_biomedicines, 2024, doi:10.3390/biomedicines12071560_

Round 1
Reviewer 1 Report
Comments and Suggestions for Authors
This review is devoted to the molecular changes in the ischemic brain as non-invasive brain stimulation targets – TMS and tDCS mechanisms, therapeutic challenges and combination therapies. This review explores the therapeutic effects of TMS and tDCS in stroke rehabili tation. A highlight is put on specific molecular mechanisms occurring in the ischemic brain and detailed molecular mechanisms of action of TMS and tDCS including the newly emerging therapeutic targets such as ferroptotic and pyroptotic pathways. Furthermore, the translational challenges of non-invasive brain stimulation (NIBS) techniques are analysed, and limitations of current clinical trials are investigated. Although more research is still necessary to link the therapeutic effects of NIBS with molecular mechanisms occurring in the ischemic brain, recent advances in non-invasive stroke treatment and growing understanding of processes involved in brain stimulation techniques provide a promising basis for finding an effective form of stroke rehabilitation method.
The article may be published after several corrections. Written in the text, that Zhou et al. demonstrated that tDCS upregulated BDNF–TrkB and its downstream PI3K/Akt signalling which antagonizes the pro-apoptotic activity of Bcl-2 family eventually protecting neurons from apoptosis [44]. In my opinion, it is necessary to specify specific proteins of Bcl-2 family, which have a pro-apoptotic function.
Author Response
Comment 1: In my opinion, it is necessary to specify specific proteins of Bcl-2 family, which have a pro-apoptotic function.
I agree with this comment, Bcl-2 family proteins have been specified, page 5, line 144, marked in red
Reviewer 2 Report
Comments and Suggestions for Authors
The review by Markowska and Tarnacka is of interest in the field. Some sentences would benefit from a language revision and a more detailed explanation. For example this sentence is quite general and is not adding any information: "Although more research is still necessary to link the therapeutic effects of NIBS with molecular mechanisms occurring in the ischemic brain, recent advances in non-invasive stroke treatment and growing understanding of processes involved in brain stimulation techniques provide a promising basis for finding an effective form of stroke rehabilitation method.".
Or: "However, they require high patient compliance and significant 41 amounts of time.": try to better specify or quantify.
Schematics or figures (also from original papers) would be of help.
Of note, some important citations are missing and should be added:
- Longo et al., 2022 showing that the rate of motor recovery is accelerated by tDCS applied in the subacute phase of stroke.
- Peruzzotti-Jametti et al., 2013 showing that tDCS exerts a measurable neuroprotective effect in the acute phase of stroke (confirmed by Notturno 2014)
- Blaschke et al., 2023 reporting network changes induced by tDCS
- Cambiaghi et al., 2021, showing the effects of rTMS on M1 (And 2022 in the mPFC)
- Liu et al., 2024 reporting that integrin α3 is required for HF rTMS-induced glutamatergic synaptic transmission
- Interesting technical aspects of rTMS can be found in Choung, J.S., Bhattacharjee, S., Son, J.P. et al. Development and application of rTMS device to murine model. Sci Rep 13, 5490 (2023).
Comments on the Quality of English LanguageThe ms would benefit from a review from a native speaker
Author Response
Comment 1:
Some sentences would benefit from a language revision and a more detailed explanation. For example this sentence is quite general and is not adding any information: "Although more research is still necessary to link the therapeutic effects of NIBS with molecular mechanisms occurring in the ischemic brain, recent advances in non-invasive stroke treatment and growing understanding of processes involved in brain stimulation techniques provide a promising basis for finding an effective form of stroke rehabilitation method.". Or: "However, they require high patient compliance and significant 41 amounts of time.": try to better specify or quantify.
I agree with the comment, theses sentences have been rephrased and specified, page 11 - lines 371-373 marked in red and page 2 - lines 40-42, marked in red
Comment 2
"Schematics or figures (also from original papers) would be of help"
I agree with this comment, I added two figures, page 9 - line 292 and page 10 - line 326
Comment 3
Of note, some important citations are missing and should be added:
- Longo et al., 2022 showing that the rate of motor recovery is accelerated by tDCS applied in the subacute phase of stroke.
- Peruzzotti-Jametti et al., 2013 showing that tDCS exerts a measurable neuroprotective effect in the acute phase of stroke (confirmed by Notturno 2014)
- Blaschke et al., 2023 reporting network changes induced by tDCS
- Cambiaghi et al., 2021, showing the effects of rTMS on M1 (And 2022 in the mPFC)
- Liu et al., 2024 reporting that integrin α3 is required for HF rTMS-induced glutamatergic synaptic transmission
- Interesting technical aspects of rTMS can be found in Choung, J.S., Bhattacharjee, S., Son, J.P. et al. Development and application of rTMS device to murine model. Sci Rep 13, 5490 (2023).
I agree with this comment, all studies have been cited in the paper, page 2 - lines 56-61, page 3 - lines 73-82, page 10 - lines 340-343, all marked in red
Round 2
Reviewer 2 Report
Comments and Suggestions for Authors
The authors have thoroughly addressed all of my concerns in the revised version, resulting in a manuscript that now comprehensively encompasses the field's literature.